# ON THE INEFFECTIVENESS OF VARIANCE REDUCED OPTIMIZATION FOR DEEP LEARNING

## ABSTRACT

The application of stochastic variance reduction to optimization has shown remarkable recent theoretical and practical success. The applicability of these techniques to the hard non-convex optimization problems encountered during training of modern deep neural networks is an open problem. We show that naive application of the SVRG technique and related approaches fail, and explore why.

## 1 INTRODUCTION

Stochastic variance reduction (SVR) consists of a collection of techniques for the minimization of finite-sum problems:

$$f(w) = \frac{1}{n} \sum_{i=1}^{N} f_i(w),$$

such as those encountered in empirical risk minimization, where each $f_i$ is the loss on a single training data point. Principle techniques include SVRG (Johnson & Zhang, 2013), SAGA (Defazio et al., 2014a), and their variants. SVR methods use control variates to reduce the variance of the traditional stochastic gradient descent (SGD) estimate $f'_i(w)$ of the full gradient $f'(w)$. Control variates are a classical technique for reducing the variance of a stochastic quantity without introducing bias. Say we have some random variable $X$. Although we could use $X$ as an estimate of $E[X] = \bar{X}$, we can often do better through the use of a control variate $Y$. If $Y$ is a random variable correlated with $X$ (i.e. $\text{Cov}[X, Y] > 0$), then we can estimate $\bar{X}$ with the quantity

$$Z = X - Y + E[Y].$$

This estimate is unbiased since $-Y$ cancels with $E[Y]$ when taking expectations, leaving $E[Z] = E[X]$. As long as $Var[Y] \le 2\text{Cov}[X, Y]$, the variance of $Z$ is lower than that of $X$.

Remarkably, these methods are able to achieve linear convergence rates for smooth strongly-convex optimization problems, a significant improvement on the sub-linear rate of SGD. SVR methods are part of a larger class of methods that explicitly exploit finite-sum structures, either by dual (SDCA, Shalev-Shwartz & Zhang, 2013; MISO, Mairal, 2014; Finito, Defazio et al., 2014b) or primal (SAG, Schmidt et al., 2017) approaches.

Recent work has seen the fusion of acceleration with variance reduction (Shalev-Shwartz & Zhang (2014); Lin et al. (2015); Defazio (2016); Allen-Zhu (2017)), and the extension of SVR approaches to general non-convex (Allen-Zhu & Hazan, 2016; Reddi et al., 2016) as well as saddle point problems (Balamurugan & Bach, 2016).

In this work we study the behavior of variance reduction methods on a prototypical non-convex problem in machine learning: A deep convolutional neural network designed for image classification. We discuss in Section 2 how standard training and modeling techniques significantly complicate the application of variance reduction methods in practice, and how to overcome some of these issues. In Sections 3 & 5 we study empirically the amount of variance reduction seen in practice on modern CNN architectures, and we quantify the properties of the network that affect the amount of variance reduction. In Sections 6 & 7 we show that streaming variants of SVRG do not improve over regular SVRG despite their theoretical ability to handle data augmentation. In Section 8 we study properties of DNN problems that actually give stochastic gradient descent an advantage over variance reduction techniques.

STANDARD SVR APPROACH

The SVRG method is the simplest of the variance reduction approaches to apply for large-scale problems, so we will focus our initial discussion on it. In SVRG, training epochs are interlaced with snapshot points where a full gradient evaluation is performed. The iterate at the snapshot point $\tilde{w}$ is stored, along with the full gradient $f'(\tilde{w})$. Snapshots can occur at any interval, although once per epoch is the most common frequency used in practice. The SGD step $w_{k+1} = w_k - \gamma f_i'(w_k)$, using the randomly sampled data-point loss $f_i$ with step size $\gamma$, is augmented with the snapshot gradient using the control variate technique to form the SVRG step:

$$w_{k+1} = w_k - \gamma \left[ f_i'(w_k) - f_i'(\tilde{w}) + f'(\tilde{w}) \right]. \tag{1}$$

The single-data point gradient $f_i'(\tilde{w})$ may be stored during the snapshot pass and retrieved, or recomputed when needed. The preference for recomputation or storage depends a lot on the computer architecture and its bottlenecks, although recomputation is typically the most practical approach.

Notice that following the control variate approach, the expected step, conditioning on $w_k$, is just a gradient step. So like SGD, it is an unbiased step. Unbiasedness is not necessary for the fast rates obtainable by SVR methods, both SAG (Schmidt et al., 2017) and Point-SAGA (Defazio, 2016) use biased steps, however biased methods are harder to analyze. Note also that successive step directions are highly correlated, as the $f'(\tilde{w})$ term appears in every consecutive step between snapshots. This kind of step correlation is also seen in momentum methods, and is considered a contributing factor to their effectiveness (Kidambi et al., 2018).

## 2 COMPLICATIONS IN PRACTICE

Modern approaches to training deep neural networks deviate significantly from the assumptions that SVR methods are traditionally analyzed under. In this section we discuss the major ways in which practice deviates from theory and how to mitigate any complications that arise.

DATA AUGMENTATION

In order to achieve state-of-the-art results in most domains, data augmentation is essential. The standard approach is to form a class of transform functions $\mathcal{T}$; for an image domain typical transforms include cropping, rotation, flipping and compositions thereof. Before the gradient calculation for a data-point $x_i$, a transform $T_i$ is sampled and the gradient is evaluated on its image $T_i(x_i)$.

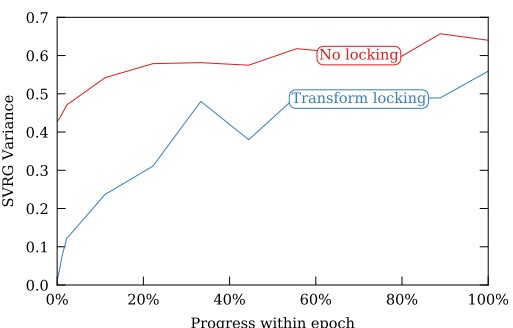

Figure 1: Variance within epoch two during LeNet training on CIFAR10.

When applying standard SVRG using gradient recomputation, the use of random transforms can destroy the prospects of any variance reduction if different transforms are used for a data-point during the snapshot pass compared to the following steps. Using a different transform is unfortunately the most natural implementation when using standard libraries (PyTorch[1]; TensorFlow, Abadi et al. (2015)), as the transform is applied automatically as part of the data-pipeline. We propose the use of transform *locking*, where the transform used during the snapshot pass is cached and reused during the following epoch/s.

This performance difference is illustrated in Figure 1, where the variance of the SVRG step is compared with and without transform locking during a single epoch during training of a LeNet model. Data augmentation consisted of random horizontal flips and random cropping to 32x32, after padding by 4 pixels on each side (following standard practice).

For SVRG with transform locking, the variance of the step is initially zero at the very beginning of the epoch, increasing over the course of the epoch. This is the behavior expected of SVRG on finite

---

[1]http://pytorch.org/

sum problems. In contrast, without transform locking the variance is non-zero at the beginning of the epoch, and uniformly worse.

The handling of data augmentation in finite-sum methods has been previously considered for the MISO method (Bietti & Mairal, 2017), which is one of the family of gradient table methods (as with the storage variant of SVRG). The stored gradients are updated with an exponential moving average instead of overwriting, which averages over multiple past transformed-data-point gradients. As we show in Section 5, stored gradients can quickly become too stale to provide useful information when training large models.

### BATCH NORMALIZATION

Batch normalization (Ioffe & Szegedy, 2015) is another technique that breaks the finite-sum structure assumption. In batch normalization, mean and variance statistics are calculated within a mini-batch, for the activations of each layer (typically before application of a nonlinearity). These statistics are used to normalize the activations. The finite sum structure no longer applies since the loss on a datapoint $i$ depends on the statistics of the mini-batch it is sampled in.

The interaction of BN with SVRG depends on if storage or recomputation of gradients is used. When recomputation is used naively, catastrophic divergence occurs in standard frameworks. The problem is a subtle interaction with the internal computation of running means and variances, for use at test time.

In order to apply batch normalization at test time, where data may not be mini-batched or may not have the same distribution as training data, it is necessary to store mean and variance information at training time for later use. The standard approach is to keep track of a exponential moving average of the mean and variances computed at each training step. For instance, PyTorch by default will update the moving average $m_{EMA}$ using the mini-batch mean $m$ as:

$$m_{EMA} = \frac{9}{10} m_{EMA} + \frac{1}{10} m.$$

During test time, the network is switched to evaluation mode using `model.eval()`, and the stored running mean and variances are then used instead of the internal mini-batch statistics for normalization. The complication with SVRG is that during training the gradient evaluations occur both at the current iterate $x_k$ and the snapshot iterate $\tilde{x}$. If the network is in train mode for both, the EMA will average over activation statistics between two different points, resulting in **poor results and divergence**.

Switching the network to evaluation mode mid-step is the obvious solution, however computing the gradient using the two different sets of normalizations results in additional introduced variance. We recommend a BN *reset* approach, where the normalization statistics are temporarily stored before the $\tilde{w}$ gradient evaluation, and the stored statistics are used to undo the updated statistics by overwriting afterwards. This avoids having to modify the batch normalization library code. It is important to use train mode during the snapshot pass as well, so that the mini-batch statistics match between the two evaluations.

### DROPOUT

Dropout (Srivastava et al., 2014) is another popular technique that affects the finite-sum assumption. When dropout is in use, a random fraction, usually 50%, of the activations will be zero at each step. This is extremely problematic when used in conjunction with variance reduction, since the sparsity pattern will be different for the snapshot evaluation of a datapoint compared to its evaluation during the epoch, resulting in much lower correlation and hence lower variance reduction.

The same dropout pattern can be used at both points as with the transform locking approach proposed above. The seed used for each data-point's sparsity pattern should be stored during the snapshot pass, and reused during the following epoch when that data-point is encountered. Storing the sparsity patterns directly is not practical as it will be many times larger than memory even for simple models.

Residual connection architectures benefit very little from dropout when batch-norm is used (He et al., 2016; Ioffe & Szegedy, 2015), and because of this we don't use dropout in the experiments detailed in this work, following standard practice.

ITERATE AVERAGING

Although it is common practice to use the last iterate of an epoch as the snapshot point for the next epoch, standard SVRG theory requires computing the snapshot at either an average iterate or a randomly chosen iterate from the epoch instead. Averaging is also needed for SGD when applied to non-convex problems. We tested both SVRG and SGD using averaging of 100%, 50% or 10% of the tail of each epoch as the starting point of the next epoch. Using a 10% tail average did result in faster initial convergence for both methods before the first step size reduction on the CIFAR10 test problem (detailed in the next section). However, this did not lead to faster convergence after the first step size reduction, and final test error was consistently worse than without averaging. For this reason we did not use iterate averaging in the experiments presented in this work.

## 3 MEASURING VARIANCE REDUCTION

To illustrate the degree of variance reduction achieved by SVRG on practical problems, we directly computed the variance of the SVRG gradient estimate, comparing it to the variance of the stochastic gradient used by SGD. To minimize noise the variance was estimated using the full dataset. The transform locking and batch norm reset techniques described above were used in order to get the most favorable performance out of SVRG.

Ratios below one indicate that variance reduction is occurring, whereas ratios around two indicate that the control variate is uncorrelated with the stochastic gradient, leading to an *increase* in variance. For SVRG to be effective we need a ratio below $1/3$ to offset the additional computational costs of the method. We plot the variance ratio at multiple points within each epoch as it changes significantly during each epoch. An initial step size of 0.1 was used, with 10-fold decreases at 150 and 220 epochs. A batch size of 128 with momentum 0.9 was used for all methods.

To highlight differences introduced by model complexity, we compared four models:

1. The classical LeNet-5 model (Lecun et al., 1998), modified to use batch-norm and ReLUs, with approximately 62 thousand parameters[2].

2. A ResNet-18 model (He et al., 2016), scaled down to match the model size of the LeNet model by halving the number of feature planes at each layer. It has approximately 69 thousand parameters.

3. A ResNet-110 model with 1.7m parameters, as used by He et al. (2016).

4. A wide DenseNet model (Huang et al., 2017) with growth rate 36 and depth 40. It has approximately 1.5 million parameters and achieves below 5% test error.

Figure 2 shows how this variance ratio depends dramatically on the model used. For the LeNet model, the SVRG step has consistently lower variance, from 4x to 2x depending on the position within the epoch, during the initial phase of convergence.

In contrast, the results for the DenseNet-40-36 model as well as the ResNet-110 model show an *increase* in variance, for the majority of each epoch, up until the first step size reduction at epoch 150. Indeed, even at only 2% progress through each epoch, the variance reduction is only a factor of 2, so computing the snapshot pass more often than once an epoch can not help during the initial phase of optimization.

The small ResNet model sits between these two extremes, showing some variance reduction mid-epoch at the early stages of optimization. Compared to the LeNet model of similar size, the modern architecture with its greater ability to fit the data also benefits less from the use of SVRG.

## 4 SNAPSHOT INTERVALS

The number of stochastic steps between snapshots has a significant effect on the practical performance of SVRG. In the classical convex theory the interval should be proportional to the condition number (Johnson & Zhang, 2013), but in practice an interval of one epoch is commonly used, and that is what we used in the experiment above. A careful examination of our results from Figure 2

---

[2]Connections between max pooling layers and convolutions are complete, as the symmetry breaking approach taken in the the original network is not implemented in modern frameworks.

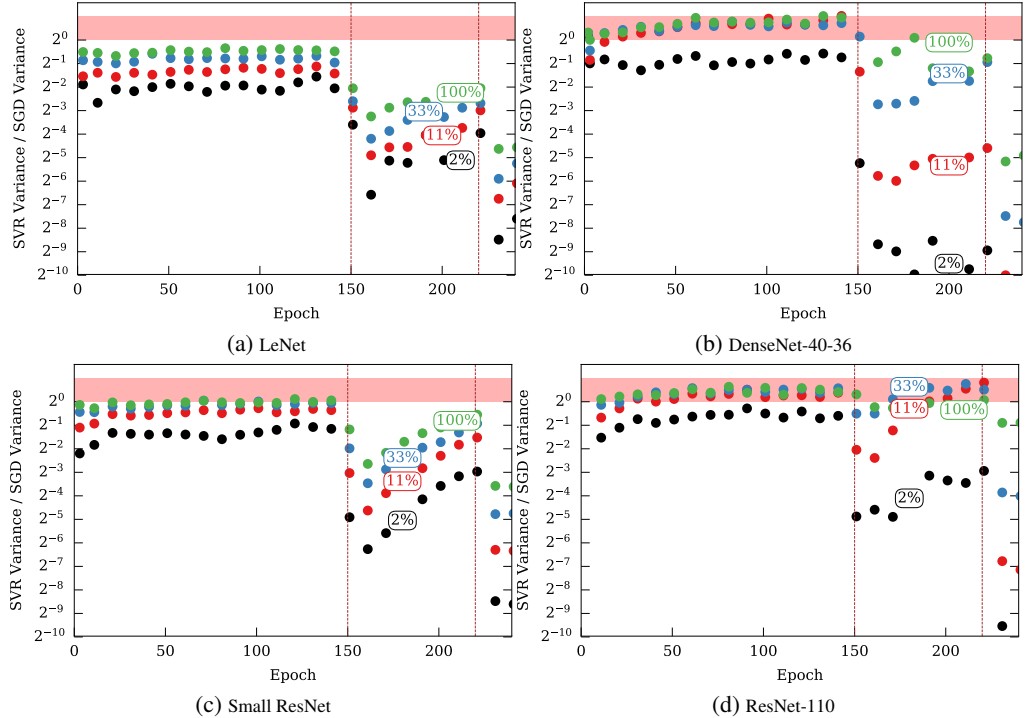

Figure 2: The SVRG to SGD gradient variance ratio during a run of SVRG. The shaded region indicates a variance *increase*, where the SVRG variance is worse than the SGD baseline. Dotted lines indicate when the step size was reduced. The variance ratio is shown at different points within each epoch, so that the 2% dots (for instance) indicate the variance at 1,000 data-points into the 50,000 datapoints consisting of the epoch. Multiple percentages within the same run are shown at equally spaced epochs.
SVRG fails to show a variance reduction for the majority of each epoch when applied to modern high-capacity networks, whereas some variance reduction is seem for smaller networks.

show that no adjustment to the snapshot interval can salvage the method. The SVRG variance can be kept reasonable (i.e. below the SGD variance) by reducing the duration between snapshots, however for the ResNet-110 and DenseNet models, even at 11% into an epoch, the SVRG step variance is already larger than that of SGD, at least during the crucial 10-150 epochs. If we were to perform snapshots at this frequency the wall-clock cost of the SVRG method would go up by an order of magnitude compared to SGD, while still under-performing on a per-epoch basis.

Similarly, we can consider performing snapshots at less frequent intervals. Our plots show that the variance of the SVRG gradient estimate will be approximately 2x the variance of the SGD estimate on the harder two problems in this case (during epochs 10-150), which certainly will not result in faster convergence. This is because the correction factor in Equation 1 becomes so out-of-date that it becomes effectively uncorrelated with the stochastic gradient, and since it's magnitude is comparable (the gradient norm decays relatively slowly during optimization for these networks) adding it to the stochastic gradient results in a doubling of the variance.

## 4.1 Variance reduction and optimization speed

For sufficiently well-behaved objective functions (such as smooth & strongly convex), we can expect that an increase of the learning rate results in a increase of the converge rate, up until the learning rate approaches a limit defined by the curvature ($\approx 1/L$ for L Lipschitz-smooth functions). This holds also in the stochastic case for small learning rates, however there is an additional ceiling that occurs as you increase the learning rate, where the variance of the gradient estimate begins to slow convergence. Which ceiling comes into effect first determines if a possible variance reduction (such

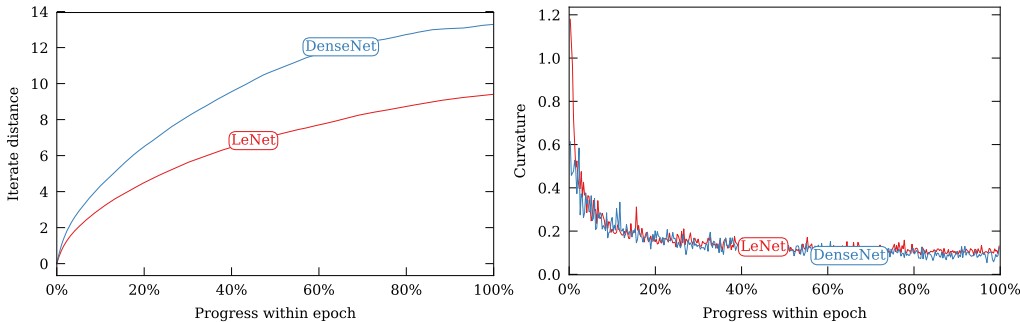

Figure 3: Distance moved from the snapshot point, and curvature relative to the snapshot point, at epoch 50.

as from SVRG) can allow for larger learning rates and thus faster convergence. Although clearly a simplified view of the non-differentiable non-convex optimization problem we are considering, it still offers some insight.

Empirically deep residual networks are known to be constrained by the curvature for a few initial epochs, and afterwards are constrained by the variance. For example, Goyal et al. (2017) show that decreasing the variance by increasing the batch-size allows them to proportionally increase the learning rate for variance reduction factors up to 30 fold. This is strong evidence that a SVR technique that results in significant variance reduction can potentially improve convergence in practice.

## 5 WHY VARIANCE REDUCTION FAILS

Figure 2 clearly illustrates that for the DenseNet model, SVRG gives no actual variance reduction for the majority of the optimization run. This also holds for larger ResNet models (plot omitted). The variance of the SVRG estimator is directly dependent on how similar the gradient is between the snapshot point $\tilde{x}$ and the current iterate $x_k$. Two phenomena may explain the differences seen here. If the $w_k$ iterate moves too quickly through the optimization landscape, the snapshot point will be too out-of-date to provide meaningful variance reduction. Alternatively, the gradient may just change more rapidly in the larger model.

Figure 3 sheds further light on this. The left plot shows how rapidly the current iterate moves within the same epoch for LeNet and DenseNet models when training using SVRG. The distance moved from the snapshot point increases significantly faster for the DenseNet model compared to the LeNet model.

In contrast the right plot shows the curvature change during an epoch, which we estimated as:

$$\left\| \frac{1}{|S_i|} \sum_{j \in S_i} \left[ f'_j(w_k) - f'_j(\tilde{w}) \right] \right\| / \left\| w_k - \tilde{w} \right\|,$$

where $S_i$ is a sampled mini-batch. This can be seen as an empirical measure of the Lipschitz smoothness constant. Surprisingly, the measured curvature is very similar for the two models, which supports the idea that iterate distance is the dominating factor in the lack of variance reduction. The curvature is highest at the beginning of an epoch because of the lack of smoothness of the objective (the Lipschitz smoothness is potentially unbounded for non-smooth functions).

Several papers have show encouraging results when using SVRG variants on small MNIST training problems (Johnson & Zhang, 2013; Lei et al., 2017). Our failure to show any improvement when using SVRG on larger problems should not be seen as a refutation of their results. Instead, we believe it shows a fundamental problem with MNIST as a baseline for optimization comparisons. Particularly with small neural network architectures, it is not representative of harder deep learning training problems.

### 5.1 SMOOTHNESS

Since known theoretical results for SVRG apply only to smooth objectives, we also computed the variance when using the ELU activation function (Clevert et al., 2016), a popular smooth activation that can be used as a drop-in replacement for ReLU. We did see a small improvement in the degree

of variance reduction when using the ELU. There was still no significant variance reduction on the DenseNet model.

# 6 STREAMING SVRG VARIANTS

In Section 3, we saw that the amount of variance reduction quickly diminished as the optimization procedure moved away from the snapshot point. One potential fix is to perform snapshots at finer intervals. To avoid incurring the cost of a full gradient evaluation at each snapshot, the class of streaming SVRG (Frostig et al., 2015; Lei et al., 2017) methods instead use a *mega-batch* to compute the snapshot point. A mega-batch is typically 10-32 times larger than a regular mini-batch. To be precise, let the mini-batch size be $b$ be and the mega-batch size be $B$. Streaming SVRG alternates between computing a snapshot mega-batch gradient $\tilde{g}$ at $\tilde{w} = w_k$, and taking a sequence of SVRG inner loop steps where a mini-batch $S_k$ is sampled, then a step is taken:

$$w_{k+1} = w_k - \gamma \left[ \frac{1}{n} \sum_{i \in S_k} \left( f_i'(w_k) - f_i'(\tilde{w}) \right) + \tilde{g} \right]. \qquad (2)$$

Although the theory suggests taking a random number of these steps, often a fixed $m$ steps is used in practice.

In this formulation the data-points from the mega-batch and subsequent $m$ steps are independent. Some further variance reduction is potentially possible by sampling the mini-batches for the inner step from the mega-batch, but at the cost of some bias. This approach has been explored as the Stochastically Controlled Stochastic Gradient (SCSG) method (Lei & Jordan, 2017).

To investigate the effectiveness of streaming SVRG methods we produced variance-over-time plots. We look at the variance of each individual step after the computation of a mega-batch, where our mega-batches were taken as 10x larger than our mini-batch size of 128 CI-FAR10 instances, and 10 inner steps were taken per snapshot. The data augmentation and batch norm reset techniques from Section 2 were used to get the lowest variance possible. The variance is estimated using the full dataset at each point.

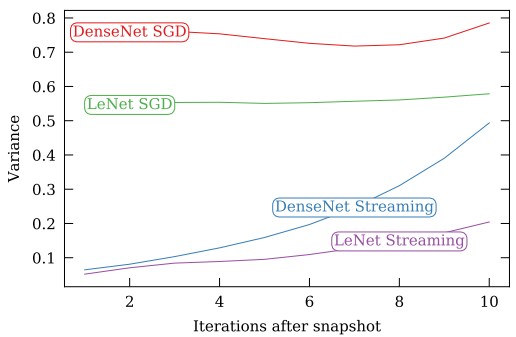

Figure 4: Streaming SVRG Variance at epoch 50

Figure 4 shows the results at the beginning of the 50th epoch. In both cases the variance is reduced by 10x for the first step, as the two mini-batch terms cancel in Equation 2, resulting in just the mega-batch being used. The variance quickly rises thereafter. These results are similar to the non-streaming SVRG method, as we see that much greater variance reduction is possible for LeNet. Recall that the amortized cost of each step is three times that of SGD, so for the DenseNet model the amount of variance reduction is not compelling.

# 7 CONVERGENCE RATE COMPARISONS

Together with the direct measures of variance reduction in Section 3, we also directly compared the convergence rate of SGD, SVRG and the streaming method SCSG. The results are shown in Figure 5. An average of 10 runs is shown for each method, using the same momentum (0.9) and learning rate (0.1) parameters for each, with a 10-fold reduction in learning rate at epochs 150 and 225. A comparison was also performed on ImageNet with a single run of each method.

The variance reduction seen in SVRG comes at the cost of the introduction of *heavy correlation between consecutive steps*. This is why the reduction in variance does not have the direct impact that increasing batch size or decreasing learning rate has on the convergence rate, and why convergence theory for VR methods requires careful proof techniques. It is for this reason that the amount of variance reduction in Figure 4 doesn't necessarily manifest as a direct improvement in convergence rate in practice. On the LeNet problem we see that SVRG converges slightly faster than SGD, whereas on the larger problems including ResNet on ImageNet (Figure 5b) and DenseNet on CIFAR10 they

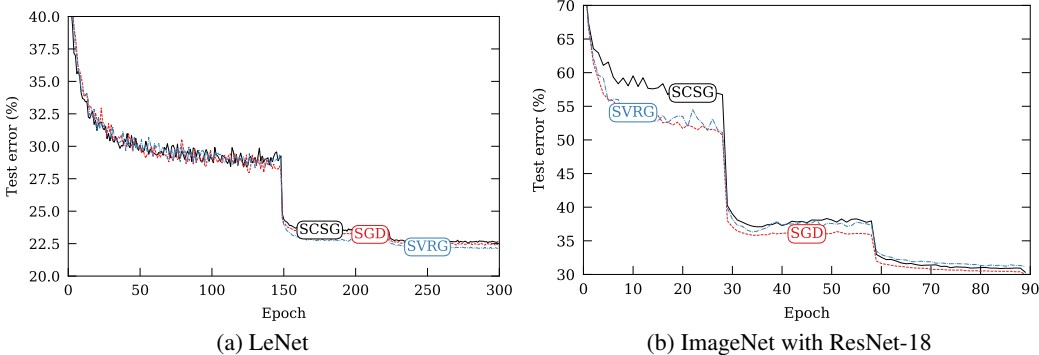

(a) LeNet                    (b) ImageNet with ResNet-18

Figure 5: Test error comparison with an average of 10 runs shown. Epochs rather than gradient evaluations are plotted on the $x$ axis. Best performing hyper-parameters are used.

| Layers | Planes | Params. | Variance | Error @ 50 epochs | Final error | Final error (SGD) |
|--------|--------|---------|----------|-------------------|-------------|-------------------|
| 20 | 16 | 0.3m | **1.80** ($\pm$.01) | 16.7% ($\pm$.2) | 8.84% ($\pm$.03) | 8.64% ($\pm$.02) |
| 56 | 16 | 0.8m | 1.83 ($\pm$.01) | 15.0% ($\pm$.2) | 7.22% ($\pm$.03) | 7.06% ($\pm$.02) |
| 92 | 16 | 1.4m | 1.88 ($\pm$.01) | **14.8%** ($\pm$.2) | 6.84% ($\pm$.02) | 6.65% ($\pm$.03) |
| 128 | 16 | 2.0m | 1.98 ($\pm$.01) | 15.1% ($\pm$.2) | **6.77%** ($\pm$.02) | 6.50% ($\pm$.02) |
| 164 | 16 | 2.6m | 2.13 ($\pm$.03) | 15.4% ($\pm$.2) | 6.84% ($\pm$.04) | **6.30%** ($\pm$.02) |
| 200 | 16 | 3.2m | 2.28 ($\pm$.03) | 16.0% ($\pm$.2) | 7.02% ($\pm$.05) | 6.54% ($\pm$.03) |

(a) Deep networks

| Layers | Planes | Params. | Variance | Error @ 50 epochs | Final error | Final error (SGD) |
|--------|--------|---------|----------|-------------------|-------------|-------------------|
| 20 | 16 | 0.3m | 1.80 ($\pm$.01) | 16.7% ($\pm$.2) | 8.84% ($\pm$.03) | 8.64% ($\pm$.02) |
| 20 | 32 | 1.1m | 1.83 ($\pm$.01) | 13.8% ($\pm$.2) | 6.74% ($\pm$.02) | 6.46% ($\pm$.03) |
| 20 | 48 | 2.4m | 1.78 ($\pm$.01) | 12.7% ($\pm$.2) | 5.97% ($\pm$.02) | 5.92% ($\pm$.02) |
| 20 | 64 | 4.3m | 1.75 ($\pm$.01) | 12.5% ($\pm$.2) | 5.69% ($\pm$.02) | 5.65% ($\pm$.02) |
| 20 | 80 | 6.8m | 1.67 ($\pm$.02) | 11.8% ($\pm$.2) | 5.46% ($\pm$.02) | 5.45% ($\pm$.02) |
| 20 | 96 | 9.7m | **1.60** ($\pm$.02) | **11.5%** ($\pm$.2) | **5.39%** ($\pm$.02) | **5.38%** ($\pm$.01) |

(b) Wide networks

Table 1: Variance and test error average when using SVRG and SGD between epochs 45-50 for a variety of ResNet architectures. Standard errors are shown using 9 runs for each architecture with different seeds. Planes is the number of activation planes after the first convolution.

are a little slower than SGD . This is consistent with the differences in the amount of variance reduction observed in the two cases in Figure 2, and our hypothesis that SVRG performs worse for larger models. The SCSG variant performs the worst in each comparison.

## 8   GRADIENT VARIANCE

A key difference between the theoretical rate for SGD and SVRG is the dependence on the variance of the gradient. SVRG's convergence rate does not depend on the variance of the gradient, whereas SGD crucially does. SVRG should perform relatively better for very high gradient variance problems, assuming the Lipschitz smoothness is comparable.

Surprisingly, we found that the gradient variance only increases modestly as depth increases for ResNet architectures. Scaling the depth of the network 10 fold (Table 1) only increases the variance $\approx 25\%$, and scaling the width actually leaves the variance roughly 10% smaller. These small changes give some indication why SVRG doesn't perform better for the larger architectures. Table 1 also shows that the test error at 50 epochs is highly correlated with the variance. The deeper models with 100+ layers actually have worse test error at epoch 50 than the baseline 20 layer model. Their higher variance results in slower convergence when using a fixed step size. In contrast, increasing the model width while fixing the number of layers results in consistently lower gradient variance as well as lower test error at epoch 50. These results suggest that lower gradient variance is a con-

tributing factor to the success of wider models such as the WRN (Zagoruyko & Komodakis, 2016). Notice also that there is a test error gap between SVRG and SGD for the fully trained deep models, whereas the wide models have no apparent gap.

## CONCLUSION

The negative results presented here are disheartening, however we don't believe that they rule out the use of stochastic variance reduction on deep learning problems. Rather, they suggest avenues for further research. For instance, SVR can be applied adaptively; or on a meta level to learning rates; or scaling matrices; and can potentially be combined with methods like Adagrad (Duchi et al., 2011) and ADAM Kingma & Ba (2014) to yield hybrid methods.

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
