# OpenReview forum: "On the Ineffectiveness of Variance Reduced Optimization for Deep Learning"
_ICLR.cc/2019/Conference_

### Official Review · AnonReviewer2 · 2018-10-15
**Thorough investigation on well-studied topic, but is it novel enough to be worthy of publication?**

**Rating:** 5
**Confidence:** 4

**Review:**

This paper is thorough and well-written. On first look, the paper seems to be addressing a topic that I believe is already well-known in the DL community that has typically been explained by memory constraints or light empirical evidence. However, a more in-depth reading of this paper shows that the authors provide a serious attempt at implementing SVRG methods. This is demonstrated by their detailed implementation that attempts to overcome the main practical algorithmic concerns for neural networks (which may be beneficial even in the implementation of other optimization algorithms for deep learning) and their in-depth experiments that give concrete evidence towards a reasonable explanation of why SVRG methods currently do not work for deep learning. In particular, they claim that because the SVRG estimator fails to significantly decrease the variance, the increased computation is not worthwhile in improving the efficiency of the algorithm. Because the empirical study is fairly in-depth and thorough and the paper itself is well-written (particularly for DL), I’m more inclined to accept the paper; however, I still do not believe that the explanation is significant and novel enough to be worthy of publication, as I will explain below.

1. Performance of SVRG methods in convex setting

It is fairly well-known that even in the convex optimization community (training logistic regression), SVRG methods fail to improve the performance of SG in the initial phase; see [1, 5]. Often, the improvements in the algorithm are seen in the later phases of the algorithm, once the iterates is sufficiently close to the solution and the linear convergence rate kicks in.

The experiments for neural networks presented here seem to corroborate this; in particular, the variance reduction introduces an additional cost but without much benefit in the main initial phase (epochs up to 150).

One also typically observes (in the convex setting) very little difference in test error between SG and SVRG methods since it is unnecessary to train logistic regression to a lower error for the test error to stabilize. Hence, the test error results in the neural network setting feels unsurprising.

2. Comments/questions on experiments

(i) Batch Normalization: When you were obtaining poor results and divergence with applying SVRG directly with training mode on, what batch size were you using for the batch normalization? In particular, did you use full batch when computing the batch norm statistics at the snapshot point? Did you try fixing the batch normalization “ghost batch size” [4]?

(ii) Measuring Variance Reduction: When training the models, what other hyperparameters were tried? Was SVRG sensitive to the choices in hyperparameters? What happened when momentum was not used? How did the training loss behave? It may also be good to mention which datasets were used since these networks have been applied to a wider set of datasets.

(iii) Streaming SVRG Variants: Have you considered SVRG with batching, as proposed in [3]?

(iv) Convergence Rate Comparisons: Is it reasonable to use the same hyperparameter settings for all of the methods? One would expect that each method needs to be tuned independently; otherwise, this may indicate that SVRG/SCSG and the SG method are so similar that they can all be treated the same as the SG method.

3. Generalization

For neural networks, the question of generalization is almost as important as finding a minimizer efficiently, which is not addressed in-depth in this paper. The SG method benefits from treating both the empirical risk and expected risk problems “equally”, whereas SVRG suffers from utilizing this finite-sum/full-batch structure, which may potentially lead to deficiencies in the testing error. In light of this, I would suggest the authors investigate more carefully the generalization properties of the solutions of SVRG methods for neural networks. This may be highly relevant to the work on large-batch training; see [6].

Summary:

Overall, the paper is quite thorough and well-written, particularly for deep learning. However, the paper still lacks enough content and novelty, in my opinion, to warrant acceptance. They appeal to a simple empirical investigation of the variance as supportive evidence for their claim; if the paper had some stronger mathematical justification specific for neural networks demonstrating why the theory does not hold in this case, the paper would be a clear accept. For these reasons, I have given a weak reject. A response addressing my concerns above and emphasizing the novelty of these results for neural networks may push me the other way.

References:
[1] Bollapragada, Raghu, et al. "A progressive batching L-BFGS method for machine learning." arXiv preprint arXiv:1802.05374(2018).
[2] Friedlander, Michael P., and Mark Schmidt. "Hybrid deterministic-stochastic methods for data fitting." SIAM Journal on Scientific Computing 34.3 (2012): A1380-A1405.
[3] Harikandeh, Reza, et al. "Stopwasting my gradients: Practical svrg." Advances in Neural Information Processing Systems. 2015.
[4] Hoffer, Elad, Itay Hubara, and Daniel Soudry. "Train longer, generalize better: closing the generalization gap in large batch training of neural networks." Advances in Neural Information Processing Systems. 2017.
[5] Johnson, Rie, and Tong Zhang. "Accelerating stochastic gradient descent using predictive variance reduction." Advances in neural information processing systems. 2013.
[6] Keskar, Nitish Shirish, et al. "On large-batch training for deep learning: Generalization gap and sharp minima." arXiv preprint arXiv:1609.04836 (2016).
[7] Smith, Samuel L., Pieter-Jan Kindermans, and Quoc V. Le. "Don't Decay the Learning Rate, Increase the Batch Size." arXiv preprint arXiv:1711.00489 (2017).

---

> ### Author Response · Authors · 2018-11-16
> **Rebuttal**
>
>
> Thank you for the detailed comments! We would like to elaborate a little on the significance of our contribution in the hopes you will change your mind.
>
> We strongly believe that a paper such as ours that points out a problem in a well-known existing method should be publishable. Currently, other researchers will end up duplicating effort in trying the promising SVRG method on deep learning problems, only to find that it doesn't work. By exploring this in depth, and pointing out how existing papers testing it on tiny networks such as LeNet are misleading, we provide a valuable contribution that can potentially spur future research into SVRG variants that can work for neural network training.
>
> We will also address specific comments of yours individually:
>
> 1) It is definitely the case that typical convex logistic regression test problems show relatively little test loss advantage when using SVRG, compared to the rapid train loss convergence. We don't believe that this is always the case though, and certainly the theory doesn't rule out faster test loss convergence, particularly if the optimization problem is badly conditioned. So it is not aprior given that SVRG should fail here.
>
> 2) i) We used batch-size 128 which is typical for these problems both at training, eval time and at the snapshot point. Ghost batch norm is recommended when training for much larger batches, and typically it is configured to have the same statistic variances as batch-size 128 or 256, so it did not seem sensible to us to use it here.
>
> ii) We did an extensive set of experiments covering learning rate, momentum and weight decay. In the end we found that SVRG behaved essentially the same as SGD with respect to these parameters. We can update the paper to include a comment to this effect if you wish.
>
> iii) We have not tried the exact prcedure described in that paper for minibatching. It looks to rely on the Lipschitz constant heavily, which is hard to work with for non-smooth optimization problems such as ReLU neural networks.
>
> iv) As mentioned in (ii), we did try other parameter settings and we found that using the same settings actually did work the best. We should definitely elaborate more on this in the camera ready, as one of the other reviewers commented on this aspect as well.
>
> 3) Generalization with SVRG is certainly an interesting research direction.

---

### Official Review · AnonReviewer1 · 2018-10-31
**Interesting investigation of the applicability of SVGD to modern neural networks**

**Rating:** 6
**Confidence:** 3

**Review:**

This work investigates the applicability of SVGD to modern neural networks and shows the naive application of SVGD typically fail. The authors find that the naive application of batch norm, dropout, and data augmentation deviate significantly from the assumptions of SVGD and variance reduction can fail easily in large nets due to the weight moves quickly in the training.

This is a thorough exploration of a well-known algorithm - SVGD in deep neural networks. Although most results indicate that SGVD fails to reduce the variance in training deep neural networks, it might still provide insights for other researchers to improve SVGD algorithm in the context of deep learning. Therefore, I'm inclined to accept this paper.

One weakness of this work is that it lacks instructive suggestion of how to improve SVGD in deep neural networks and no solution of variance reduction is given.

Finally, I'd like to pose a question: Is it really useful to reduce variance in training deep neural networks? We've proposed tons of regularizers to increase the stochasticity of the gradient (e.g., small-batch training, dropout, Gaussian noise), which have been shown to improve the generalization. I agree optimization is important, but generalization is arguably the ultimate goal for most tasks.

---

> ### Author Response · Authors · 2018-11-14
> **Rebuttal**
>
> The question you pose is an interesting one. It is definitely the case that additional gradient noise is helpful right at the very beginning of optimization. In practice this phase where the noise is useful lasts for only a few epochs, as has been shown recently in a series of papers (e.g. https://arxiv.org/abs/1706.02677) applying large-batch (8k or greater) to imagenet. The general result is that smaller batches (i.e higher variance gradients) help final validation set error when used for the first ~5 epochs, after which much larger batches can be used without harming generalization performance. We believe that these large-batch training results show quite clearly that there is significant gains to be had with variance reduction, as most of the training time is spent in the remaining 85 epochs.

---

> > ### Comment · AnonReviewer1 · 2018-12-04
> > **I stick to my rating**
> >
> > The authors gave a serious attempt at experimenting SVGD with modern neural networks and show the naive application of SVGD typically fails. I really appreciate their thorough investigation of SVGD, though there's no solution proposed to improve SVGD. So I'm inclined to accept this paper and keep my rating unchanged.

---

### Official Review · AnonReviewer3 · 2018-11-02
**Reasonable idea**

**Rating:** 5
**Confidence:** 5

**Review:**

This paper presents an analysis of SVRG style methods, which have shown remarkable progress in improving rates of convergence (in theory) for convex and non-convex optimization (Reddi et al 2016).

This paper highlights some of the difficulties faced by SVRG in practice, especially for practically relevant deep learning problems. Issues such as dropout, batch norm, data augmentation (random crop/rotation/translations) tend to cause additional issues with regards to increasing bias (and/or variance) to resulting updates. Some fixes are proposed by this paper in order to remove these issues and then reconsider the resulting algorithm's behavior in practice. These fixes appear right headed and the observation about ratio of variances (of stochastic gradients) with or without variance reduction is an interesting way to track benefits of variance reduction.

There are some issues that I'd like to raise in the paper's consideration of variance reduction:

[1] I'd like more clarification (using an expression or two) to make sure I have the right understanding of the variance estimates computed by the authors for variance reduced stochastic gradient and the routine stochastic gradient that is computed.

[2] In any case, the claim that if the ratio of the variance of the gradient computed with/without variance reduction is less than 1/3, thats when effective variance reduction is happening is true only in the regime when the variance (estimation error) dominates the bias (approximation error). At the start of the optimization, the bias is the dominating factor variance reduction isn't really useful. That gets us to the point below.

[3] It is also important to note that variance reduction really doesn't matter at the initial stages of learning. This is noticed by the paper, which says that variance reduction doesn't matter when the iterates move rather quickly through the optimization landscape - which is the case when we are at the start of the optimization. In fact, performing SGD over the first few passes/until the error for SGD starts "bouncing around'' is a very good idea that is recommended in practice (for ex., see the SDCA paper of Shalev-Shwarz and Zhang (2013)). Only when the variance of SGD's iterates starts dominating the initial error, one requires to use one of several possible alternatives, including (a) decaying learning rate or, (b) increasing batch size or, (c) iterate averaging or, (d) variance reduction.  Note that, in a similar spirit, Reddi et al. (2016) mention that SVRG is more sensitive to the initial point than SGD for non-convex problems.


With these issues in mind, I'd be interested in understanding how variance reduction behaves for all networks once we start at an epoch when SGD's iterates start bouncing around (i.e. the error flattens out). Basically, start out with SGD with a certain step size until the error starts bouncing around, then, switch to SVRG with all the fixes (or without these fixes) proposed by the paper. This will ensure that the variance dominates the error and this is where variance reduction should really matter. Without this comparison, while this paper's results and thoughts are somewhat interesting, the results are inconclusive.

---

> ### Author Response · Authors · 2018-11-14
> **Rebuttal**
>
>
> Thank you for providing detailed comments. We will respond to your comments individually.
>
> 1) Yes, we can provide more detail here in our camera ready, although your understanding sounds correct.
>
> 2/3) We agree completely with your characterization of the error. Variance reduction can certainly not help when variance is not the limiting factor. However, for problems such as imagenet, there is strong evidence to suggest that variance is the limiting factor for the majority of the optimization time. For instance, the seminal imagenet-in-1-hour paper (https://arxiv.org/abs/1706.02677) established that the learning rate can be increased if the batchsize is also proportionally increased, but only after a warm-up period of 5-10 epochs. This shows that variance rather than curvature is the limiting factor, and that the majority of training occurs in this variance constrained situation.
>
> The fact that the learning rate is also commonly decreased at 30 epochs of 90 suggests (although doesn't prove) that variance is the dominating factor by that point, as learning rate decreases primarily used to decrease variance.
>
> 4) Regarding the technique of switching to SVRG after using SGD in a warm-up phase, we did investigate this thoroughly in unreported experiments. In no situation were we able to get a non-trivial improvement. Switching to SVRG 160+ epochs into CIFAR-10 training is a viable technique, particularly for the smaller LeNet, however it doesn't result in significant convergence rate improvements as the majority of the optimization time is spent in the earlier phases, and the variance reduction for DenseNet/ResNet-100 is too small even at those later stages, as shown in our primary plots in Figure 2.
>
> The amount of variance reduction shown in Figure 2 at 160+ epochs is similar to the variance reduction you would see from using SGD for the earlier epochs then switching over.

---

> > ### Comment · AnonReviewer3 · 2018-11-29
> > **response**
> >
> > Thanks for the response.
> >
> > The fact that the imagenet training in one hour paper suggests that learning rate \propto batchsize after 5-10 epochs is not significant of variance issues in optimization. Note that at the start of the optimization, there is no variance. The variance accumulates as a stochastic algorithm is run for longer. The first 5-10 epochs are just significant of the rather uneven and unpredictable landscape of the objective at the start - this is a function of the initialization, because, if we "initialized" our model at the iterate offered by the optimization algorithm after 10 epochs, clearly every issue such as learning rate \propto batchsize etc actually begins to hold and issues representing bias > variance (at the start of the optimization) also begins to hold.
> >
> > Again, please note that the phase between 10 and 90 epochs is actually the point where the bias is still a dominating factor over the variance. The first 10 epochs can be considered as the phase where we try to escape the rather poor landscape of the loss function that is a consequence of our initialization.
> >
> > With regards to experiments on variance reduction: great that you have run these experiments. One point to note is the following: at least in theory, variance reduction doesn't require the learning rate to be decayed. Firstly, note that the variance reduction phase must be begun when the error levels off, which is at 90 epochs (according to your reply). Secondly, the learning rate (as per theory) must be held fixed. Starting variance reduction with a super small learning rate is uncalled for because the variance existing in SGD's iterates is already sufficient and doesn't require explicit variance reduction tricks (variance reduction doesn't really benefit here). As a summary, my prescription is the following: run SGD until the error levels off (which is roughly around 90 epochs as I read from your comments). Keep the learning rate constant and turn on the variance reduction at this phase. Compare this with SGD with some tuning of the learning rate. I would really believe this should compete fairly well with SGD, otherwise, there is something about results such as Reddi et al. (2016) that don't apply to practical non-convex optimization (which is rather puzzling to me).

---

> > > ### Author Response · Authors · 2018-11-29
> > > **Further response**
> > >
> > > The need to reduce the step-size seems in practice is another example of how the SVRG theory fails for realistic deep learning problems, due to the broken assumptions we discuss in the paper. You mention "otherwise, there is something about results such as Reddi et al. (2016) that don't apply to practical non-convex optimization", this is exactly the purpose of our paper; to discuss in detail the assumptions that are broken, and show that plausible fixes don't work!
> > >
> > > The idea of switching over to SVRG after using SGD for 90 epochs or more is a sound idea in theory. We have actually run extensive experiments to this end. The issue is that the majority of the computation time of learning will have already been spent on the earlier epochs, so only tiny improvements in the total wall-clock time of the whole run can be expected. Recall that we show in Figure 2 that the amount of variance reduction achieved in the best case at the later stages of optimization is very small, at least for large neural networks. This makes large improvements simply impossible, at least using existing techniques.
> > > We hope that we have answered your questions to your satisfaction. Please let us know if we should make changes to the paper to address any of these questions, and if that would help raise your rating of the paper.

---

> > > > ### Comment · AnonReviewer3 · 2018-11-30
> > > > **response**
> > > >
> > > > I think my comment has been misconstrued. I mentioned the following: begin variance reduction at the 90th epoch with the same learning rate that has been used until the 90th epoch. This is because the theory (and behavior for the (strongly) convex cases) indicate that the learning rate should be held a constant when variance reduction is used. Turning the learning rate down along with variance reduction is wasteful, because once the learning rate is turned down, the variance in SGD's iterates also reduces. The real impact of variance reduction is that it allows the use of constant learning rates through the course of optimization.
> > > >
> > > > The question I ask is the following: Have you experimented with beginning the variance reduction at the 90th epoch while keeping the learning rate a constant, or say cut the learning rate by some small value like 2/3 instead of a factor of 10? How does this compare to SGD with the same learning rate used through the course of optimization? How does this compare to SGD with learning rate cut by a constant factor every time the error plateaus?
> > > >
> > > > At some level, I wish to see experiments and plots of the sort described in the previous paragraph, because I believe that is needed to draw conclusions and consider changes to the score.

---

> > > > > ### Author Response · Authors · 2018-11-30
> > > > > **further response**
> > > > >
> > > > > Thank you for clarifying. I have run experiments using smaller learning rate reductions than 1/10, as I was thinking along the same lines as you suggest. Essentially, the thought was that if significant variance reduction is occurring, then it should be possible to reduce the learning rate by a smaller amount to get the same gradient variance at each stage of optimization.
> > > > > In practice I was only able to get this to work for the smallest of the neural networks I tried, as the amount of variance reduction that occurs with the larger networks is too small.
> > > > >
> > > > > Unfortunately we are past the deadline for paper revisions, so OpenReview won't allow me to modify the submitted paper at add additional plots. I will add plots showing this effect to the camera ready based on your suggestion if the paper is accepted.
> > > > >
> > > > > Thanks for the detailed feedback!

---

### Public Comment · (anonymous) · 2018-12-05
**Inspiring work, variance reductions up to date**

The author's work seems trying to solve a very important dilemma: the mismatch of theory and practice on variance reduction. I appreciate the work very much, and would like to bring two recent papers I found:

1. SPIDER: Near-Optimal Non-Convex Optimization via Stochastic Path Integrated Differential Estimator (NIPS 2018, https://arxiv.org/abs/1807.01695)
2. SpiderBoost: A Class of Faster Variance-reduced Algorithms for Nonconvex Optimization (https://arxiv.org/abs/1810.10690)

Paper 1 is highly related to SARAH (Nguyen et al ICML 2017) and improves the variance reduction complexity to O(n^{1/2} \epsilon^{-2}) \land O(\epsilon^{-3}), which is theoretically better than most variance reduction papers mentioned in this paper and is easy to implement. Paper 2 improves Paper 1 and relaxes the learning rate condition while maintaining the same convergence rate, which is known to be near-optimal. There is also a concurrent papar [Zhou, Xu, Gu NIPS 2018 http://arxiv.org/abs/1806.07811] which achieves the same rate but significantly more complex.

It might be just my personal interest to test whether these new theoretically optimal algorithms would be truly accelerating DL training in practice, in additional to the authors' experimental findings, and I would strongly recommend the authors to take a look or even try them out and include the results into your comparison table in your follow-ups.

---

### Meta-Review · Area_Chair1 · 2018-12-18
**Revise and resubmit**

**Confidence:** 4
**Recommendation:** Reject

**Metareview:**

All reviewers agreed that this paper addresses an important question in deep learning (why doesn't SVRG help for deep learning)? But the paper still has some issues that need to be addressed before publication, thus the AC recommends "revise and resubmit".